# 650 W All-Fiber Single-Frequency Polarization-Maintaining Fiber Amplifier Based on Hybrid Wavelength Pumping and Tapered Yb-Doped Fibers

Wanpeng Jiang [1,2], Changsheng Yang [2,3,4,5,*], Qilai Zhao [1,2], Quan Gu [2,3], Jiamin Huang [2,3], Kui Jiang [2,3], Kaijun Zhou [2], Zhouming Feng [1,2,5], Zhongmin Yang [1,2,4,6] and Shanhui Xu [1,2,3,4,5,6,7,*]

1   School of Physics and Optoelectronics, South China University of Technology, Guangzhou 510640, China; 202020130464@mail.scut.edu.cn (W.J.); zhaoql@scut.edu.cn (Q.Z.); zhmfeng@scut.edu.cn (Z.F.); yangzm@scut.edu.cn (Z.Y.)
2   State Key Laboratory of Luminescent Materials and Devices and Institute of Optical Communication Materials, South China University of Technology, Guangzhou 510640, China; 201921021032@mail.scut.edu.cn (Q.G.); jmhuang0925@foxmail.com (J.H.); 202021021733@mail.scut.edu.cn (K.J.); stephenzhou@lightpromotech.com (K.Z.)
3   School of Materials of Science and Engineering, South China University of Technology, Guangzhou 510640, China
4   Guangdong Engineering Technology Research and Development Center of Special Optical Fiber Materials and Devices, Guangzhou 510640, China
5   Guangdong Engineering Technology Research and Development Center of High-Performance Fiber Laser Techniques and Equipments, Zhuhai 519031, China
6   Guangdong Provincial Key Laboratory of Fiber laser Materials and Applied Techniques, South China University of Technology, Guangzhou 510640, China
7   Hengqin Firay Sci-Tech Company Ltd., Zhuhai 519031, China
*   Correspondence: mscsyang@scut.edu.cn (C.Y.); flxshy@scut.edu.cn (S.X.)

**Abstract:** Based on hybrid wavelength pumping and tapered Yb-doped fibers (T-YDFs), a 650 W all-fiber single-frequency polarization-maintaining fiber amplifier was demonstrated experimentally at 1030 nm. Different pump power ratios in the T-YDF-based power-amplifier stage were proposed to investigate their influence on the transverse mode instability (TMI) effect. The highest TMI threshold was obtained when the pump power ratio of 940 nm to 976 nm was 1:4.4. A measured $M^2$ factor of 1.7 and a polarization extinction ratio of 14 dB at the maximum output power were obtained. To the best of our knowledge, these results exhibit the highest output power of any all-fiber single-frequency polarization-maintaining fiber amplifiers created up to now.

**Keywords:** single frequency; high power; tapered Yb-doped fiber; hybrid wavelength pumping

## 1. Introduction

Research and development into high-power single-frequency fiber lasers (SFFLs) has continued up to the present day. Due to their compact structure [1], narrow linewidth [2], and high beam quality [3,4], they are widely used in nonlinear frequency conversion [5], coherent beam combining [6], and gravitational wave detection (GWD) [7]. Especially in the field of GWD, they show a higher output power and higher detection sensitivity in their measuring systems [7]. With the promotion of the master oscillator power amplifier (MOPA) structure, fiber amplifiers have become an active component in improvements in the output power of SFFLs. However, the main limitation of further power scaling for single-frequency fiber amplifiers (SFFAs) is the onset of detrimental effects in the optical fibers, such as stimulated Brillouin scattering (SBS) and transverse mode instability (TMI) [8,9]. For SBS suppression, the most common method to combat this is to employ an active fiber with a large mode area (LMA) or high pump absorption [10]. Besides this, some suppression

methods for broadening the Brillouin gain spectrum have also been proposed, such as applying strain or a temperature gradient [11,12].

To date, SFFAs with an output power of hundreds of watts have been demonstrated successfully. In 2013, Zhang et al. achieved an output power of 170 W using a 10/125 μm core/inner-cladding diameter polarization-maintaining (PM) Yb-doped fiber with an applied strain gradient method [13]. In 2014, Huang et al. achieved a 414 W single-frequency PM laser output using a 25/250 μm core/inner-cladding diameter PM Yb-doped LMA fiber, applying a step-distributed longitudinal strain gradient [14]. In the same year, Robin et al. achieved the highest output power record yet of 811 W in an SFFA via a bulk optic configuration based on a Yb-doped PM photonic crystal fiber and a thermal gradient technique [15]. Evidently, power amplification for all-fiber SFFAs is still restricted by the onset of SBS with relatively small core diameters of ≤25 μm, while applying active fibers with larger mode field areas will easily introduce the TMI effect as a new challenge [16].

To balance these two limiting factors and to achieve a higher power diffraction-limited output, tapered Yb-doped fibers (T-YDFs) have become the first choice due to their inherent superior characteristics: their relatively simple structure and their good compatibility with other optical components [17,18]. For these fibers, their mode field area increases along the tapered region, leading to the gradual strengthening of SBS suppression. Moreover, the depressed-cladding design of T-YDFs enhances high-order mode (HOM) filtering, which contributes to the maintenance of a high-quality beam output. In 2020, Lai et al. achieved an output power of 550 W at 1030 nm by employing a T-YDF, which is at present the highest output power of any all-fiber single-frequency amplifier, with an $M^2$ factor of 1.47 [19]. Higher output powers are limited by the occurrence of the TMI effect; particularly for high-power fiber amplifiers, photodarkening and quantum defects will bring heat loads to the active fiber, which accelerates the occurrence of TMI—leading to beam quality degradation [20,21]. Consequently, numerous techniques have been proposed to suppress the TMI effect, such as hybrid wavelength pumping methods, adopting a smaller winding radius for active fibers to filter out HOMs, pumping structure optimization [21,22], and so on.

Thanks to a large emission cross-section and the low level of quantum defects in Yb-doped fibers at 1030 nm [23], fiber laser working in this wavelength has the potential to excite a higher output power in a fiber laser system. In this paper, a 650 W all-fiber single-frequency PM fiber amplifier is demonstrated experimentally at 1030 nm. With the combination of a T-YDF and a hybrid-pumping scheme, the SBS was suppressed and the TMI threshold was enhanced successfully. Moreover, the measured $M^2$ factor and the polarization extinction ratio (PER) were 1.7 and 14 dB at the maximum output power, respectively.

## 2. Experimental Setup

The experimental setup of the all-fiber single-frequency PM fiber amplifier is shown in Figure 1. A MOPA structure was applied, containing a 1030 nm linearly polarized single-frequency seed laser, two pre-amplifiers, and a power-amplifier. The seed laser had an output power of 15 mW and a laser linewidth of 13 kHz. More details on the single-frequency seed laser can be found in our previous works [24,25]. The signal from the seed laser was first boosted to 9 W through two pre-amplifiers (both of them shared a similar amplification configuration). In the two pre-amplifier stages, two pieces of active fiber (2.5 m-long PM-YDF-10/125 and 3.5 m-long PM-YDF-12/125) were forward pumped by 976 nm multi-mode laser diodes (MM-LDs) via (2 + 1) × 1 PM combiners. Two customized integrated optical components (PM-ISO-BPF) acted as PM isolators, and band-pass filters were used. An 1/99 2 × 2 PM coupler was employed to monitor the backward propagating lights via its 1% port for the evaluation of SBS.

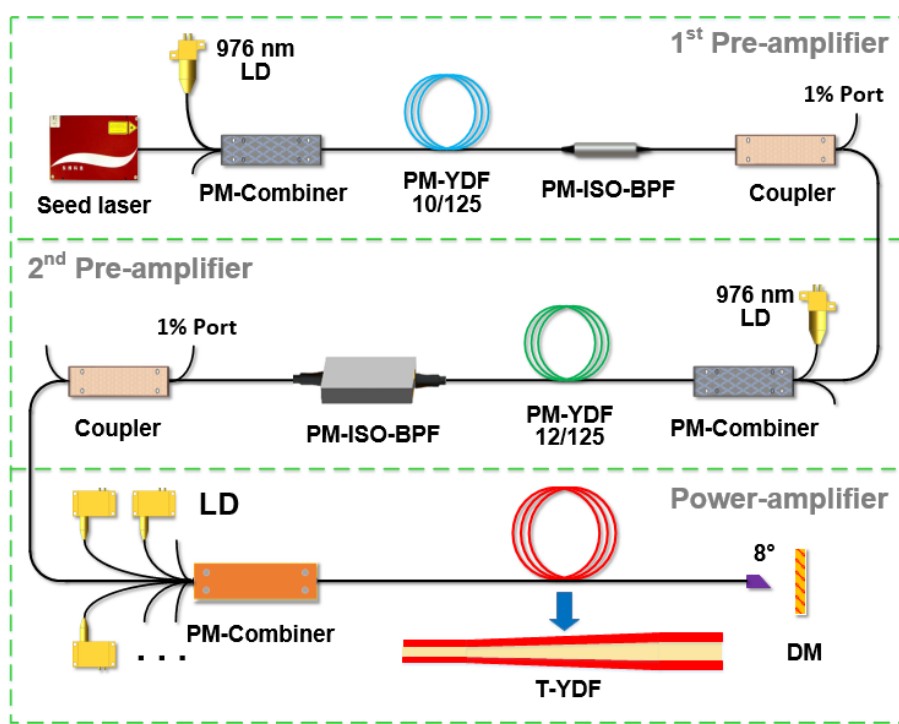

**Figure 1.** Experimental setup of the all-fiber single-frequency PM fiber amplifier. (PM: polarization-maintaining, LD: laser diode, YDF: Yb-doped fiber, BPF: band-pass filter, ISO: isolator, T-YDF: tapered Yb-doped fiber, DM: dichroic mirror).

In the power-amplifier stage, six 130 W 976 nm MM-LDs and two 150 W 940 nm MM-LDs are optional for providing pump power via a $(6 + 1) \times 1$ PM combiner. For this combiner, the core/inner-cladding diameter of the input and output port were 10/125 μm and 25/250 μm, respectively. A 2.5 m-long T-YDF with an input and output port of 35/250 μm and 56/400 μm was used. Different from the concave profile along such fibers [17,18], the tapered region—with a length of 0.7 m—was located at the middle part of the fiber, with a linearly longitudinal profile. The cladding absorption of the T-YDF was 10 dB/m at 976 nm, and the core numerical aperture was 0.07. The T-YDF were coiled in a racetrack shape with a minimum diameter of ~8 cm and placed on a water-cooled aluminum plate with a cooling temperature of 20 °C. A 0.3 m-long non-PM passive fiber with a core/inner-diameter of 50/400 μm was spliced behind the output port of the T-YDF. The signal laser was delivered from the passive fiber with an angle 8° to prevent any optical reflections. A dichroic mirror was used to filter the residual pump power.

## 3. Results and Discussion

Due to the special structure and relatively large core-diameter of T-YDFs, a high SBS threshold can be maintained in the fiber laser system. Therefore, in the process of realizing a sub-kW level SFFA, TMI suppression became our main research goal. For Yb-doped fibers, the typical absorption peak wavelengths are located at 915 nm, 940 nm, and 976 nm. Although the cladding absorption of T-YDFs at 915 nm is about the same as at 940 nm, a high-power SFFA pumped by the latter wavelength has a lower quantum loss, which helps to mitigate the thermal effect. Thus, LDs with a central wavelength of 940 nm were adopted in the experiments. To explore the influence on the TMI of hybrid wavelength pumping, three schemes with different pumping power ratios were employed and the TMI threshold was measured separately.

### 3.1. Single Wavelength Pumping of 976 nm LDs

In this pumping scheme, six 976 nm LDs were employed for single-wavelength pumping. The output power and backward propagating power of the SFFA versus the

pump power are shown in Figure 2a. It can be seen that the output power increased linearly without any power roll-over. As the pump power increased to 718 W, the maximum output power of 603 W was obtained, with a corresponding slope efficiency of 84%. Thanks to the longer gain fiber, the SFFA was able to achieve stronger pump absorption and improve the slope efficiency. Meanwhile, the excess heat was evenly distributed throughout the relatively long fiber, which effectively increased the TMI threshold and achieved a higher power output compared with the results in Ref. [19]. Furthermore, owing to the relatively large core diameter of the T-YDFs in the power-amplifier, nonlinear increments in the backward propagating power were not observed—proving that the SBS effect did not emerge.

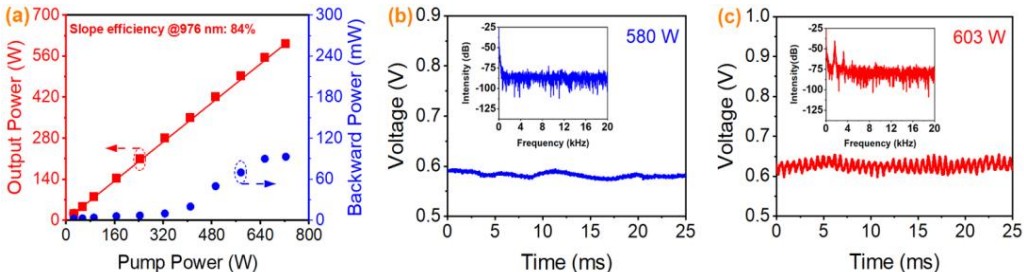

**Figure 2.** (**a**) Output power and backward propagating power versus pump power. Signals in temporal intensity. Inset: corresponding Fourier spectra. (**b**) At 580 W. (**c**) At 603 W.

It is clear that the SFFA still worked under the SBS threshold, but the TMI effect caused by the heat accumulation gradually strengthened with increases in the output power. The time-domain signal was monitored by applying a high-speed photoelectric detector (PD) whose response frequency was nearly 17 MHz displayed in the oscilloscope. The temporal intensity during a period of 25 ms and the corresponding frequency-domain signal within the frequency of 0––20 kHz by fast Fourier transform at the output power at 580 W and 603 W are illustrated in Figure 2b, Figure 2c, and their inset, respectively. The figure shows that the temporal intensity fluctuates were kept stable, and the characteristic peaks of kHz in the frequency-domain signal were not observed when the output power increased to 580 W. However, when the output power reached 603 W, an unexpected fluctuation appeared and corresponding discrete frequency peaks were observed within 0––5 kHz in the Fourier spectrum. All the results show that the TMI effect onset at a power scaling of 603 W.

### 3.2. Hybrid Pumping with a Power Ratio of 1:4.4

In the second scheme, a 940 nm LD and five 976 nm LDs were used in the hybrid pumping method. The output power of the hybrid pumping versus the pump power are depicted in Figure 3a. With the 940 nm LD module fully turned on (a pump power of 147 W), the output power increased linearly to 92 W, with a slope efficiency of 59%. Furthermore, the maximum output power of 650 W was achieved at a total pump power of 797 W, and the slope efficiency was slightly improved to 86% by 976 nm LD pumping. According to Ref. [26], the slope efficiency can be improved by hybrid pumping, which corresponds with the outcome of this study. Meanwhile, Figure 3b and its inset show the growth trend of the backward propagating power and backward spectra within 1029–1031 nm using an optical spectrum analyzer (OSA) with a resolution of 0.02 nm, indicating that the SBS effect did not occur.

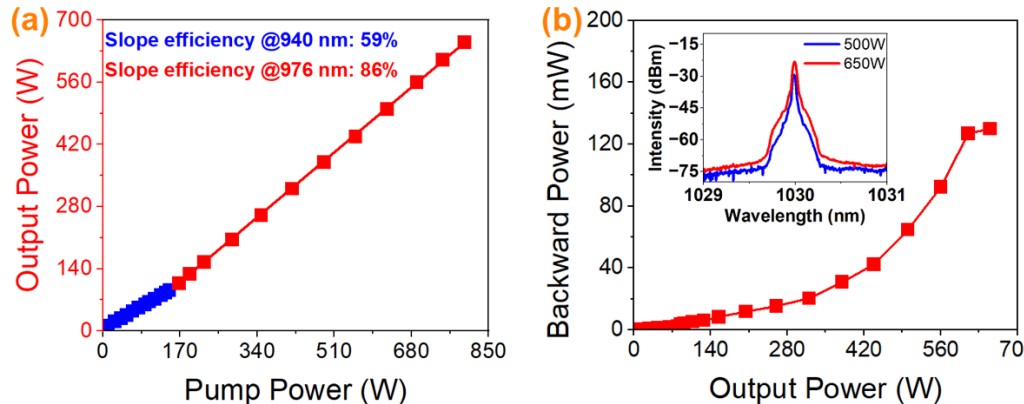

**Figure 3.** (**a**) Output power versus pump power. (**b**) Backward propagating power versus output power. Inset: Backward spectra within the wavelength range of 1029–1031 nm.

Usually, the lower pump absorption of high-power fiber amplifiers (under 940 nm LDs pumping) can enhance gain saturation and improve the TMI threshold [27]. To verify the effects of hybrid pumping on the TMI effect, the time-domain signal and its Fourier transform spectra were recorded in the process of power amplification, as shown in Figure 4a,b and its inset. The temporal intensity remained subtly fluctuating and the corresponding frequency-domain spectrum was also stable until the output power reached 628 W. However, with the power increasing to 650 W, dramatic instability appeared in the temporal trace, and corresponding frequency spectral peaks within the frequency of 0–−5 kHz could be observed easily. For comparison, the temporal intensity and the magnitude of the characteristic kHz peaks were stronger than the situation in the first system. As the output power was higher, the mode coupling effect was stronger in this SFFA. Once the output power breaks through the TMI threshold, its characteristic signal will be more obvious than at a low power output [20,21]. Therefore, the results show that the TMI threshold was increased by nearly 50 W in this laser system.

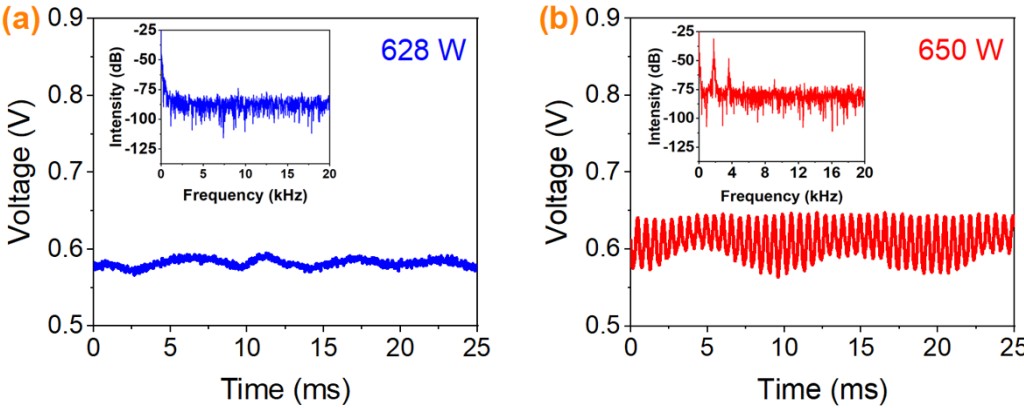

**Figure 4.** Signals for temporal intensity. Inset: corresponding Fourier spectra. (**a**) At 628 W. (**b**) At 650 W.

### 3.3. Hybrid Pumping with a Power Ratio of 1:1.7

Owing to the excellent functioning of the hybrid pumping in the previous scheme, the pump power ratio (adding a number of 940 nm LDs) was increased in this experiment. Two 940 nm LDs and four 976 nm LDs were adopted in the second hybrid pumping method. The output power and backward propagating power versus the pump power are demonstrated in Figure 5a. It is easy to find that as the 940 nm pump power reached 277 W, the output power increased to 167 W, with a corresponding slope efficiency of 59%. Then, the maximum output power of 560 W was obtained, with a slope efficiency of 82% under a 976 nm pump power of 482 W. Compared with the former systems, the slope efficiency

was not improved by hybrid pumping, and was even slightly reduced. Meanwhile, there was no sign of nonlinear growth in the backward propagating power during the power amplification, which proves that SBS did not occur in this system.

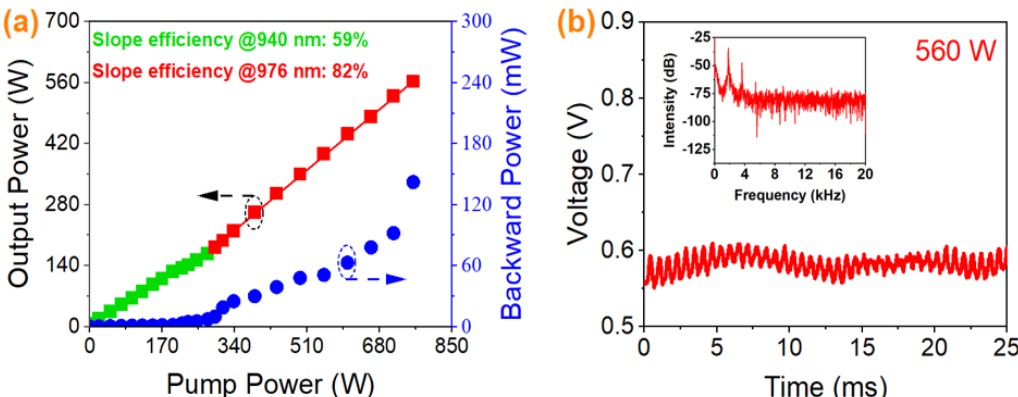

**Figure 5.** (**a**) Output power and backward propagating power versus pump power. (**b**) Signals in temporal intensity. Inset: corresponding Fourier spectra at 560 W.

Obviously, in the third system, the low optical–optical conversion efficiency of the 940 nm LD pumping contributed to excessive heat accumulation in the T-YDF, which may be a possible reason for the reduction in slope efficiency pumped by the 976 nm LDs. Meanwhile, too much heat will inevitably lead to a threshold reduction in the TMI. To further determine the TMI threshold of the new system, the temporal intensity of the output laser and the corresponding frequency-domain signal at 560 W was measured, as shown in Figure 5b and its inset. Severe fluctuations showed up in the time-domain signal and some characteristic peaks within 0–5 kHz were observed in the corresponding Fourier spectrum. It is not hard to ascertain that the TMI effect onset at 560 W, and all the results prove that the TMI threshold was reduced by 90 W compared with the second scheme.

### 3.4. Analysis and Comparison

The $M^2$ factor and PER under single wavelength pumping and the hybrid pumping (a pump power ratio of 940 nm to 976 nm is 1:1.7) are shown in Figure 6a,b and its inset, respectively. The red points in Figure 6 represent the power beyond the TMI threshold. It could be found that when the power exceeded the threshold of the TMI and increased to 593 W and 638 W in these two schemes, beam profile distortion could be observed and the $M^2$ factor increased to be 2.2 and 1.9, respectively. Meanwhile, the PER of the signal light could still be kept stable above 14 dB at the maximum power output.

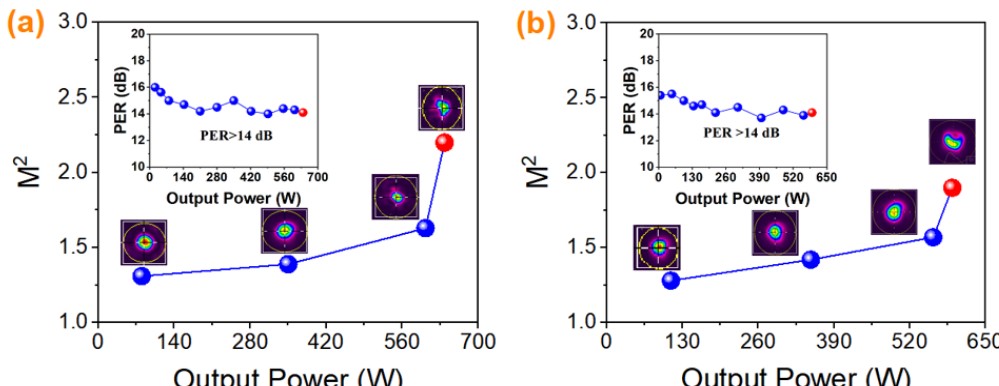

**Figure 6.** Measured beam quality and beam profile at different output powers. Inset: measured PER at different output power. (**a**) SFFA under single wavelength pumping. (**b**) SFFA under the hybrid pumping with a power ratio of 1:1.7.

By comparing the latter two hybrid wavelength pumping schemes, it could be found that balancing the gain saturation and thermal effect became the goal for optimizing the ratio of the two different pump powers. There is an optimal power ratio in the hybrid wavelength pumping scheme. Finally, in this system, the highest TMI threshold was reached at 650 W, when the pump power ratio of 940 nm to 976 nm was 1:4.4. Besides this, the $M^2$ factor and their corresponding beam profiles at different output powers are shown in Figure 7a. Obvious beam quality degradation was found when the output power increased from 320 W to 650 W and the $M^2$ factor was degraded from 1.28 to 1.7. This beam quality degradation could be attributed to the coupling process of the fundamental mode and high-order mode at high-power outputs. The PER was measured at different output powers, as displayed in Figure 7b. It can be seen that the PER of the output laser could be maintained at 14 dB, showing that the polarization degree was stable at all power scales—even though TMI had occurred. In the inset graphic, the output optical spectrum at 650 W was measured by an OSA with a resolution of 0.02 nm. The central wavelength was located at 1030 nm and the optical signal-to-noise ratio was close to 50 dB. A further study on TMI suppression based on T-YDFs will be considered as the key goal of our subsequent work.

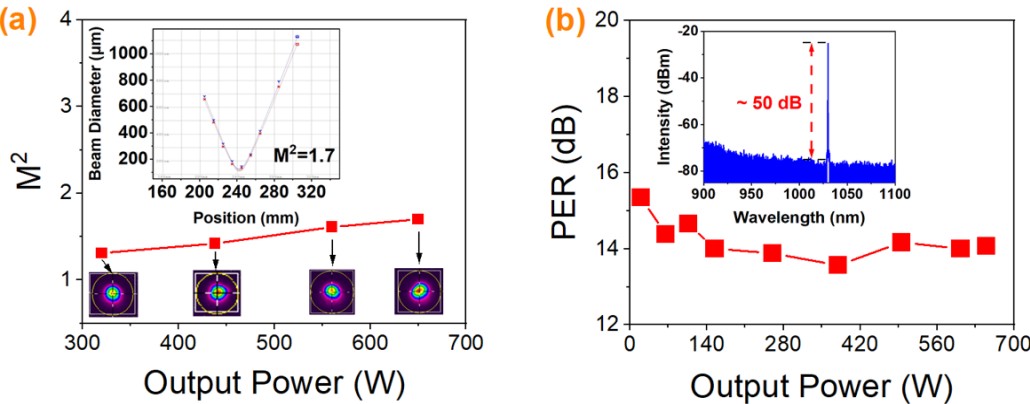

**Figure 7.** SFFA under hybrid pumping with a power ratio of 1:4.4. (**a**) Measured beam quality and beam profiles at different output powers. Inset: beam quality measurement results at 650 W. (**b**) Measured PER at different output powers. Inset: output optical spectrum at the maximum power within the wavelength range of 900–1100 nm.

Moreover, the linewidth of the output laser was measured by a delayed self-heterodyne method, which could be recorded by applying a highspeed PD and displayed in the spectrum analyzer with a resolution bandwidth of 1 kHz. Figure 8a shows the measured linewidth results of the seed laser and the SFFA under hybrid pumping with a power ratio of 1:4.4. The typical heterodyne signal was fit to the Lorentzian profile for better estimation of the spectral linewidth, which it was 264 kHz at −20 dB from the peak, indicating that the linewidth of the SFFA at 650 W was consistent with that of the seed laser—both of which were 13.2 kHz full-width at half maximum. Linewidth broadening was not observed during the power amplification, which results from the lack of an evident ASE in the optical spectrum. In Figure 8b, the single-frequency output of the SFFA was shown using a scanning spectrum of a Fabry–Perot interferometer with a free spectral range of 1.5 GHz, due to the outstanding performance of the seed laser. The absence of any peaks between the main resonance of the interferometer proved that the output laser operated stably with a single-frequency behavior, without mode hopping or mode competition phenomena.

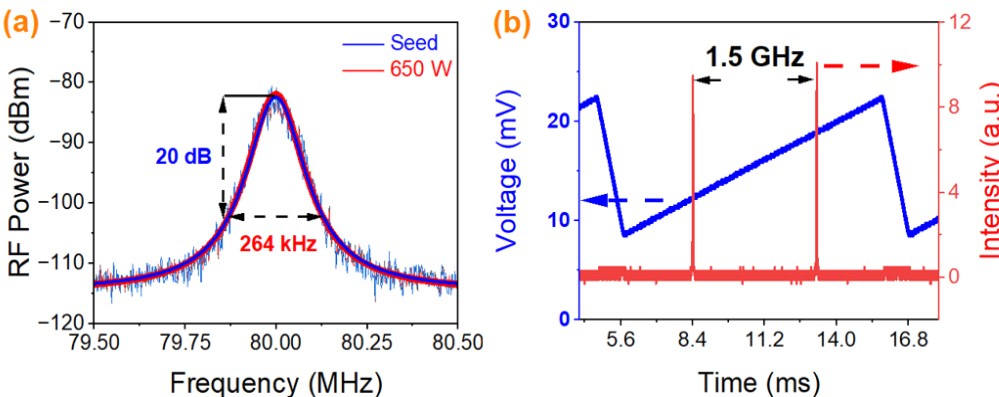

**Figure 8.** SFFA under hybrid pumping with a power ratio of 1:4.4. (**a**) Measured spectral linewidth of the seed laser and power amplifier at 650 W. (**b**) Measured longitudinal mode characteristics of the power amplifier at maximum output power.

## 4. Conclusions

In conclusion, a 1030 nm all-fiber single-frequency polarization−maintaining fiber amplifier with an output power of 650 W was demonstrated experimentally. By optimizing the pump power ratio of 940 nm to 976 nm to 1:4.4, the highest TMI threshold was realized. At the maximum output power, the $M^2$ factor was measured to be 1.7 and the PER was 14 dB, respectively. For this high-power SFFA, hybrid pumping was able to improve the TMI threshold; with the high SBS threshold of T-YDFs, the optimal parameter was obtained to achieve the highest output power in the system, which provided a possible direction for the development of high-brightness kW-class fiber amplifiers.

**Author Contributions:** Conceptualization, C.Y. and W.J.; methodology, S.X., Z.Y. and C.Y.; validation, W.J., C.Y., K.Z. and Z.F.; formal analysis, W.J., Q.Z. and C.Y.; investigation, W.J., Q.G. and K.J.; resources, W.J., C.Y. and S.X.; data curation, Q.Z. and J.H.; writing—original draft preparation, W.J.; writing—review and editing, C.Y.; visualization, W.J. All authors have read and agreed to the published version of the manuscript.

**Funding:** This research was funded by the Key-Area Research and Development Program of Guangdong Province (2018B090904001, 2018B090904003, and 2020B090922006), Major Program of the National Natural Science Foundation of China (61790582), NSFC (62035015), Fundamental Research Funds for the Central Universities (2020CG03 and 2020ZYGXZR073), Leading talents of science and technology innovation of Guangdong Special Support Plan Program (2019TX05Z344), China Postdoctoral Science Foundation (2021M101256), Guangdong Basic and Applied Basic Research Foundation (2022A1515012594), Local Innovative and Research Teams Project of Guangdong Pearl River Talents Program (2017BT01X137), and Independent Research Project of State Key Lab of Luminescent Materials and Devices, South China University of Technology (Skllmd-2022-13).

**Institutional Review Board Statement:** Not applicable.

**Informed Consent Statement:** Not applicable.

**Data Availability Statement:** The data presented in this study are available upon request from the corresponding author. The data are not publicly available due to privacy.

**Conflicts of Interest:** The authors declare no conflict of interest.

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
