# Peer review of "650 W All-Fiber Single-Frequency Polarization-Maintaining Fiber Amplifier Based on Hybrid Wavelength Pumping and Tapered Yb-Doped Fibers"

_photonics, doi:10.3390/photonics9080518_

Round 1

Reviewer 1 Report

The authors demonstrate all fiber single frequency fiber amplifier using pumping at different wavelengths. They study the performance of amplifiers with different power ratio. A few comments below:

1. The slope efficiency of 940nm LD pumping is around 59% which is much lower than that at 976nm.Could the authors comment why? 

2. What power ratio is used for Fig. 6. If it is 1:4.4, could the authors also provide similar figures for other power ratios? 

3.Could the authors comments more about why the best performance could be obtained at the power ratio of 1:4.4? It is not well explained why it should be the best.  

4. What is the Linewidth of the output? It is better to provide the linewidth spectrum for demonstrating the single frequency operation.      

Author Response

We are grateful for your insightful comments on the manuscript. Following your advice, our point-to-point response please see the attachment.

Reviewer 2 Report

Manuscript "650 W all-fiber single-frequency polarization-maintaining fiber amplifier based on hybrid wavelength pumping and tapered Yb-doped fiber" after Wanpeng Jiang et all. considers the issues of increasing the maximal output power of a single-frequency ytterbium fiber amplifier based on a tapered active optical fiber. The possibility of increasing the threshold  of transverse mode instability effect due to the combined use of pump laser diodes at two different wavelengths: 940 and 976 nm is experimentally studied. The positive effect of such pumping has been demonstrated, and the optimal ratio between the pump powers at the indicated wavelengths has been tentatively determined. 

I believe that this article is of interest to engineers and researchers working in the field of fiber lasers and amplifiers. And I think that it can be published in the journal Photonics without significant changes.

It is necessary to correct one technical defect in the text: Fig. 6 is placed on one page, and the caption to it is on the next.

Author Response

We are grateful for your comments on the manuscript. And we are very sorry for the inconvenience caused to you by our negligence. Following your advice, we have made some changes to the revised manuscript.

Reviewer 3 Report

No comments.

Author Response

We are grateful for your review of the manuscript, and express our most sincere thanks to you.

Round 2

Reviewer 1 Report

It looks better.